# Gut Microbiota and Critically Ill Patients: Immunity and Its Modulation via Probiotics and Immunonutrition

**DOI:** 10.3390/nu15163569

**Published:** 2023-08-13

**Authors:** Ludovico Abenavoli, Emidio Scarpellini, Maria Rosaria Paravati, Giuseppe Guido Maria Scarlata, Luigi Boccuto, Bruno Tilocca, Paola Roncada, Francesco Luzza

**Affiliations:** 1Department of Health Sciences, University “Magna Graecia”, 88100 Catanzaro, Italy; mrparavati@unicz.it (M.R.P.); giuseppeguidomaria.scarlata@unicz.it (G.G.M.S.); tilocca@unicz.it (B.T.); roncada@unicz.it (P.R.); luzza@unicz.it (F.L.); 2Translationeel Onderzoek van Gastro-Enterologische Aandoeningen (T.A.R.G.I.D.), Gasthuisberg University 11 Hospital, KU Leuven, Herestraat 49, 3000 Leuven, Belgium; emidio.scarpellini@med.kuleuven.be; 3School of Nursing, Healthcare Genetics Program, Clemson University, Clemson, SC 29634, USA; lboccut@clemson.edu; 4School of Health Research, Clemson University, Clemson, SC 29634, USA

**Keywords:** intensive care, gut microbiota, inflammation, probiotics, antibiotics, nutrition

## Abstract

Critically ill patients have a hyper-inflammatory response against various offending injuries that can result in tissue damage, organ failure, and fatal prognosis. The origin of this detrimental, uncontrolled inflammatory cascade can be found also within our gut. In detail, one of the main actors is our gut microbiota with its imbalance, namely gut dysbiosis: learning about the microbiota’s dysfunction and pathophysiology in the frame of critical patients is of crucial and emerging importance in the management of the systemic inflammatory response syndrome (SIRS) and the multiple organ dysfunction syndrome (MODS). Multiple pieces of evidence indicate that the bacteria that populate our gut efficiently modulate the immune response. Treatment and pretreatment with probiotics have shown promising preliminary results to attenuate systemic inflammation, especially in postoperative infections and ventilation performance. Finally, it is emerging how immunonutrition may exert a possible impact on the health status of patients in intensive care. Thus, this manuscript reviews evidence from the literature on gut microbiota composition, its derangement in critically ill patients, its pathophysiological role, and the described and emerging opportunities arising from its modulation.

## 1. Introduction

Patients affected by critical illnesses develop a potentially lethal body reaction to damage (e.g., infection, trauma, burning). The tissue damage may worsen up to the point where it causes deterioration of the organs’ function and eventually leads to their failure. Thus, the scythe of death comes closer and closer. For example, almost 50 million patients with sepsis are reported throughout the world, with a mortality of about 20% [1]. In addition, trauma is the leading cause of death in younger age [2,3]. These conditions ignite a physiological innate and adaptive immune systems response, able to manage pathogens proliferation across tissues, organs, and the whole organism. In detail, some main players of this systemic inflammatory response to damage have been described: pathogen-associated pattern molecules (PAMPS) or damage-associated pattern molecules (DAMPS). These are involved in the systemic inflammatory response syndrome (SIRS), the multiple organ dysfunction syndrome (MODS), and the related compensatory anti-inflammatory response syndrome (CAIRS) [3,4]. Indeed, prolonged SIRS suppresses Th1-type immune and increases Th2-type responses. Altogether, both features of SIRS can lead to persistent inflammation and immunosuppression. A major side effect of robust Persistent Inflammation, Immunosuppression, and Catabolism Syndrome (PICS) results in protein consumption [5]. The issue of excessive inflammatory response in critical patients can be solved through its modulation. Anti-inflammatory approaches (e.g., immunosuppressors, steroids, non-steroidal anti-inflammatory drugs) are feasible such as those described in the “cytokines’ storm” of COVID-19 [6]. However, very often these remedies are not sufficient to reach the goal of extinguishing the fire of inflammation in critical patients.

Another effective approach is to look at other items responsible for starting and maintaining inflammation in these patients. Gut dysfunction encompasses several damages offending our intestine in critically ill patients: the intestinal epithelium, intestinal immune system, and, finally, gut microbiota [3,7]. A qualitative and quantitative alteration of our gut microbiota asset, namely “dysbiosis”, can significantly affect local and systemic inflammatory response and, eventually, host immunity [8]. Gut dysbiosis and its metabolites (e.g., short-chain fatty acids (SCFAs) and trimethylamine *N*- oxide) are also associated with SIRS until the patient’s demise [9].

Thus, this review describes changes in gut microbiota composition, and metabolism in critically ill patients, and their role in prognosis and treatment. In detail, we will review evidence on gut microbiota re-establishment via the use of probiotics, symbiotics, and immunonutrition.

## 2. Methods

A narrative review of the literature was performed through NCBI PubMed and Scopus search engines. Mesh terms were the keywords: “intensive care; gut microbiota; inflammation; probiotics; antibiotics; nutrition”. The search included English papers published in each period. All types of papers were included, i.e., reviews, retrospective analyses, and experimental studies [10].

## 3. Gut Dysbiosis as the Origin of Systemic Inflammatory Response to Acute Critical Damages

Gut microbiota can be defined as a complex community populated not merely by bacteria but also yeasts, fungi, archaea, protozoa, and viruses [11]. Its main functions are involved in food digestion and absorption, immune system regulation, and metabolism development [12]. In physiological conditions, gut microbiota encompasses more than 500–1000 different bacterial species, mainly anaerobes: *Bacteroides* and *Firmicutes* are the most prevalent ones [10,13].

Under critical conditions, the composition of gut microbiota is deranged by several factors, also including treatments in use [3,10]. Among these, we must mention medications affecting gastrointestinal motility, (broad)-spectrum antibiotics, and mechanical ventilation [3,10]. In brief, we can summarize the bacterial shift typical of critically ill patients by about 10,000 times lower total anaerobes abundance (namely, *Bifidobacterium* and *Lactobacillus)* and 100 times higher *Staphylococcus* abundance vs. healthy subjects [14].

It is not easy to assess the impact of a single treatment in use in critical patients on gut dysbiosis. Indeed, two small observational studies report the reduction of microbial diversity and, on the opposite, the overgrowth of pathogenic microorganisms such as *Enterococcus* after prolonged antibiotics’ administration [15,16].

Nutrition is another factor able to contribute to gut dysbiosis in ICU patients. According to the general and neurologic conditions of the admitted patient, the availability of food nutrients via the enteral or parenteral route of administration can affect gut microbiome composition [17]. Further, skin decontamination with medications like chlorhexidine, and invasive procedures (e.g., endotracheal intubation, intravascular catheters placement, emergency bronchoscopy, GI endoscopy for bleeding, obstruction) may alter the natural human body barrier mechanisms and facilitate the access and proliferation of detrimental microbes within the gut of patients [18].

Finally, hypoperfusion and reperfusion of the intestinal wall of the critical patient can result in mucosal inflammation, increased intestinal permeability, increased nitrate concentrations, and reduced oxygen concentration [19,20]. The latter favor the growth of microbes in the *Proteobacteria* phylum that includes several gram-negative rods (namely, *Pseudomonas aeruginosa* and *Escherichia coli*), and some members of the *Firmicutes* phylum, such as *Staphylococcus aureus* and *Enterococcus* spp. [21,22].

In addition, looking at the metabolic asset of a “critically ill” dysbiotic gut microbiota, we must record that total organic acids, acetic acid, and butyric acid are significantly decreased as compared to healthy subjects (Table 1). In particular, butyric acid is significantly depleted, and fecal pH rises within the gut of critically ill patients vs. healthy controls. Thus, it is not surprising to find the number of total obligate anaerobes and of total facultative anaerobes to be significantly associated with both sepsis and related mortality in critical patients [23,24].

More recently, metagenomic analysis using the 16S rRNA gene fragments helps to reconstruct the proportion of bacterial abundance instead of their concentration [25]. In healthy subjects, *Firmicutes* or *Bacteroidetes* were predominant at the phylum level, but the ratio of *Bacteroidetes* to *Firmicutes* (B/F ratio) was associated with higher mortality in intensive care unit (ICU) patients [26]. The shift started within the first 6 h of admission at the ICU [27]. Further, Ojima et al. reported that the gut microbiota composition of patients under broad-spectrum antibiotics dramatically changed within the first week [28].

The gut is physiologically and pathophysiologically connected with other organs through several “axis”. Therefore, when our gut is hit, this event can affect other organs and result in multiple organ failure (MOF) [2,3]. Thus, this paradigm teaches us that gut dysfunction can affect the prognosis of patients suffering from sepsis, shock, trauma, burning, and bleeding. Already in 1994, Moore et al. used a reperfusion model in the rat following occlusion of the supra-mesenteric artery for 45 min and subsequent intraperitoneal injection of gram-negative lipopolysaccharide. This model showed increased white blood cell concentration, increased pulmonary permeability, and bacterial translocation with a subsequent increase in inflammation [29]. In a mouse burn model, *Escherichia coli* was found in the spleen and liver, already five minutes after injury [30]. Again, gut-derived inflammatory cytokines were found in the mesenteric lymph nodes in a rat shock model [31]. Interestingly, intestinal tight junction proteins (namely, claudin-5 and occluding) concentration significantly decreased with higher intestinal permeability in a cecal ligation puncture model [32]. Altogether, these pieces of the evidence puzzle endorse the model linking direct or indirect gut injury to the other organs.

From a clinical evidence point of view, bacterial translocation of obligate anaerobes (namely, *Clostridium coccoides* and *Bacteroides fragilis* groups) was found in the mesenteric lymph nodes of 15% of patients upon laparotomy [33] and 35% of patients after hepatectomy for biliary cancer [14].

## 4. Gut Microbiota and Immune System Disequilibrium in Critical Patient

The human gut microbiota has a particular tolerance towards our organism, guaranteed by immune system cross-talk. This tolerance looks like the mutualistic interplay of bugs’ and our metabolic processes [34]. In this milieu, obligate anaerobic bacteria are the main inhibitors of bacterial overgrowth, realizing a “colonization resistance” [35]. In addition, the relationship between gut microbiota and our immune system recognizes molecules that interact also with our metabolism [36]. Germ-free mice have thinner mucus barrier, fewer Peyer’s patches, and, again, thinner *lamina propria*. From a cellular point of view, this is associated with a decreased T cells and B cells count vs. healthy mice [37]. Further, from a molecular point of view, the immune and metabolic signals from bugs are resembled by SCFAs, coupled with pathogen recognition receptors (PPAR) (namely, Toll-like receptors and nucleotide oligomerization domain (NOD)-like receptors (NLRs)) within the gut. This milieu works out both in a healthy and a disease status in humans. If gut microbiota is depleted there is reduced myeloid-cell development in the bone marrow and delayed clearance of bacterial infection.

In the critical patient body, the adaptive immunity is altered. At the bench, Ivanov et al. showed that segmented filamentous bacteria are able to induce the differentiation of Th17 cells [38]. Moreover, Atarashi et al. showed how 17 species of bacteria, including *Clostridiales*, can induce regulatory T cells [39]. Interestingly, in ICU patients there are significantly lower proportions of *Clostridia* class bacteria vs. healthy subjects [40]. More interestingly, 11 species of bacteria belonging to the phyla *Bacteroidetes*, *Firmicutes*, and *Fusobacterium* induced IFN-producing CD8 + cells [41]. Thus, the gut microbiota has an immune-modulatory action.

Data from autopsies of septic patients showed that the abundance of CD4 +, CD8 + T cells, and HLA-DR+ cells was significantly decreased while levels of PD-1, regulatory T, and myeloid-derived suppressor cells were increased [42]. Looking at summarizing paradigms, the extreme balance of Bacteroidetes and Firmicutes ratio can be used as a predictor of mortality in ICU patients, also describing the imbalanced immunity. Furthermore, once this balance is disrupted, pathogenic bacteria might trigger an inflammatory reaction, or, a “dysbiotic” gut microbiota might further weaken the innate immune response, resulting in prolonged immunosuppression [20,21,28]. For example, seventy-one mechanically ventilated and treated with broad-spectrum antibiotics patients were enrolled, and their fecal samples were collected on days 1–2, 3–4, 5–7, and 8–14 upon ICU admission. These samples were profiled through 16S rRNA gene deep sequencing to define the microbiological composition. Importantly, five major fecal bacterial phyla were sufficiently different in each patient at admission. Interestingly, *Bacteroidetes* and *Firmicutes* abundance stabilized within the first week upon ICU admission, with a significant reduction in α-diversity. Moreover, a reduced ratio of *Bacteroidetes* to *Firmicutes* occurring within one week from admission was associated with higher mortality. Specifically, with a ratio higher than 8 or < smaller than 1/8 (odds ratio: 5.54, 95% CI: 1.39–22.18, *p* = 0.015) [35].

In this study, it is shown that both antibiotics and the severity of critical illness are associated with gut dysbiosis and *Actinobacteria* are more sensitive than other phyla to these causative factors. However, it is important to recognize that the dysbiotic ratio of *Bacteroidetes* to *Firmicutes* derangement occurred independently of the antibiotics’ administration. Other observational studies are reporting the same B/F ratio shift being associated with higher mortality in critical patients (namely, those undergoing allogenic hematopoietic cell transplantation or in neurological patients) [43,44].

## 5. Gut Dysbiosis Modulation through Probiotics and Immunonutrition

### 5.1. Probiotics and Their Derivatives

#### 5.1.1. Probiotics Definition and Effects

Probiotics are defined as alive microorganisms beneficially affecting the host [11,45]. Probiotics, which are mainly represented by *Lactobacillus* and *Bifidobacterium*, have been extensively used to treat and/or prevent antibiotic-induced diarrhea, acute infectious diarrhea, and necrotizing enterocolitis [32]. Mechanistically, probiotics are able to improve gut health and functioning through improved barrier function, namely intestinal permeability, and immune system-modulation. The latter depends on both direct probiotics’ action and their metabolic/wall components [46]. Direct effects of probiotics action pass through microorganism-associated molecular patterns (MAMPs) and pattern recognition receptors (PRRs) interaction within the gut mucosa [47,48]. At a molecular level, MAMPs examples are flagellin, lipopolysaccharide, lipoteichoic acid, and peptidoglycan. Flagellins belonging to the probiotic *E. coli* Nissle 1917 can induce the expression of beta-defensin via Toll-like receptor 5 [49]. Further, peptidoglycan translocases from the intestinal lumen to the blood torrent and increases the killing capacity of neutrophils via NOD1 [50].

Bifidobacteria are able to produce a high concentration of SCFA and, in detail, acetic acid resulting in reduced intestinal pH, increased rate of epithelium trophism, and healing in an enterohemorrhagic *E. coli* 0157 mouse model [51]. In fact, *Bifidobacterium breve* abundance has a significant correlation with tight junction-related genes [37]. Furthermore, *L. casei* increases lung epithelium-located natural killer cell activity [52].

#### 5.1.2. Probiotics and Diarrhea

In a mouse model of antibiotic-associated diarrhea (namely, clindamycin), *Clostridium butyricum* significantly decreased levels of inflammatory cytokines (e.g., IL-6, IFN-gamma) within the colon [53]. *Lactobacillus rhamnosus GG* administration increased regulatory T cells in a mouse model of pseudomonas pneumonia [54]. Interestingly, these mice had better survival vs. non-supplemented ones.

As diarrhea can significantly prolong ICU stay and increase mortality rate in ICU patients [55], Hickson et al. reported that 135 hospital patients treated with antibiotics and, as an add-on group, multistrain probiotic (namely, *L. casei*, *L. bulgaricus*, and *Streptococcus thermophiles*) had an incidence of diarrhea of 12% vs. 34% from the control group [56]. A subsequent metanalysis showed that *Saccharomyces boulardii*, *L. rhamnosus GG*, or *B. longum* are able to prevent diarrhea outbreaks in hospitalized patients [57]. Interestingly, in the ICU evidence are similar. In detail, Bleichner et al. studied 128 ICU patients and their number of days with diarrhea. The latter was significantly reduced in the *S. boulardii* group only [58]. The use of probiotics and/or symbiotics as preventive measures helps maintain gut “eubiosis” and, importantly, reduces the incidence of diarrhea [3].

#### 5.1.3. Fecal Microbiota Transplantation and Critically Ill Patient

For refractory diarrhea, there is an extreme ratio of treatment that generally is curative: fecal microbiota transplantation (FMT) [59]. The latter is the cornerstone of curative therapy for patients with multiple recurrences of *Clostridium difficile* infection (namely, CDI) [60]. In a case report of untreatable massive diarrhea associated with antibiotics, FMT led to a significant reduction in symptoms and promoted intestinal eubiosis [61]. In critically ill patients, CDI is responsible for 15–25% of nosocomial antibiotic-associated diarrhea [62]. Indeed, critically ill patients are pathophysiologically different from non-critical admitted patients. Thus, more than promising safety and efficacy data on FMT in non-critically ill humans cannot be directly translated to the ICU ones. Comorbidities such as diabetes mellitus, IBD, liver cirrhosis, chronic kidney disease (CKD), and malignancies but also gastrointestinal surgery [63,64] and increased exposure to drugs (e.g., antibiotics, immunosuppressants, PPIs, H2 blockers, NSAIDs) make the latter more vulnerable to developing CDI [52,53]. In detail, patients with fulminant colitis and/or septic shock refractory to conservative treatment have the recommendation for colectomy [53]. However, the latter is an invasive procedure with a 50% mortality rate [53]. Therefore, patients without an absolute indication for surgery (e.g., those with colonic perforation) can benefit from such as FMT [51].

There is only one randomized controlled trial on FMT for fulminant/severe CDI in critically ill patients, and data from four retrospective case-cohort studies and case reports and series. These data, all together, show FMT in these patients being feasible and with a significant reduction in mortality and morbidity vs. antibiotic therapy alone (namely, a primary cure rate of 78% and 88% of patients able to avoid surgery) [63,64]. Moreover, FMT seems to be sufficiently safe in ICU patients also.

Some RCTs are showing an encouraging statistically significant improvement of active IBD is also in the critically ill patients cohort treated with FMT vs. placebo [65].

Some case reports and two case series on patients with septic shock in the frame of severe diarrhea in the ICU treated with FMT cannot justify solid evidence supporting the use of this remedy in these patients [66,67]. Surprisingly, FMT has a very safe and effective profile of action in critically ill immunocompromised patients (namely, HIV/AIDS, hematologic malignancies, or patients undergoing immune suppressive therapy) [68,69].

Finally, evidence from an uncontrolled study on immune-compromised hematologic patients demonstrated a total elimination of multi-drug resistant organisms (MDRO)s detected in the feces of 75% of patients after FMT [70]. However, these promising results have not been confirmed in a subsequent RCT [71].

#### 5.1.4. Probiotics and Infections

Several meta-analyses have shown probiotics to reduce infectious complications of surgical procedures [72] and in trauma patients [73]. This increase in immune activity and, in particular, immune surveillance towards pathogens has been demonstrated by Sugawara et al.: patients taking probiotics upon hepatectomy, showed significantly higher natural killer cell activity, increased lymphocyte counts, and lower pro-inflammatory cytokines levels (namely, IL-6) after hepatectomy vs. control group [74].

In addition, the combination of probiotics’ strain and food components able to favor their proliferation, namely, symbiotics, are an effective measure to reduce the incidence of diarrhea and ventilator-associated pneumonia (VAP) in patients with sepsis [75]. From a meta-analysis point of view, Batra et al. found that ventilated critically ill ICU patients administered with probiotics had a reduced incidence of VAP. Of mention, the authors described also the duration of mechanical ventilation, length of ICU stay, and in-hospital mortality to be significantly reduced [76].

Looking at very tough and hard times for our humanity, we must mention gut microbiota and the COVID-19 pandemic. In these patients, there is gut dysbiosis characterized by a reduced abundance of some bacteria (namely, *Eubacterium ventricosum*, *Faecalibacterium prausnitzii*, *Reseburia*, and *Lachnospiraceae)* and an increase of potentially opportunistic strains (namely, *Clostridium hathewayi*, *Actinomyces viscosus*, and *B. nordii)* [77]. In 200 adults with severe COVID-19 pneumonia, different strains of probiotics showed a significant association with reduced risk of death [78]. However, these data were from a retrospective cohort of patients in the study.

Altogether, this evidence supports the hypothesis that probiotics and postbiotics (namely, parts of them) can modulate and, also, prevent systemic inflammation in the clinical acute setup.

#### 5.1.5. Limitations to Probiotics’ Use in ICU Patients

However, there are some limitations and, also, open questions about the use and misuse of probiotics in the ICU environment: What are the patients, especially in the ICU setting, we can administer therapies able to re-modulate gut microbiota? Are there limitations?

Intestinal ischemia, acute pancreatitis, use of opioids as analgesics, use of adrenergic drugs, massive use of fluids and, pre-existing diseases such as diabetes, neurologic disease, previous GI surgery, inflammatory bowel disease are all variables potentially affecting and limiting the use of probiotics and/or immunonutrition in ICU patients and, finally, re-establishing eubiosis [79]. Thus, intestinal functioning failure featuring impaired intestinal absorption of food nutrients and liquids, dysmotility, and, finally, increased gut permeability and related gut dysbiosis can lead to and favor SIRS development [3,54]. Therefore, GI failure happens very frequently in ICU patients, namely in about 60% of them. Subsequently, enteral nutrition is stopped in about 15% of critically ill patients [80].

For example, the gastrointestinal dysmotility typical of acute pancreatitis is significantly associated with feeding intolerance and, subsequent bacterial translocation through altered intestinal permeability [81]. Importantly, Olah et al. found the incidence of infectious complications in pancreatitis patients to be significantly reduced upon *Lactobacillus plantarum* supplementation (4.5% vs. 30.4%, *p* < 0.05) [82]. Indeed, results are not uniform but no case of administered bacterial strain translocation, potentially leading to bacteriemia and sepsis risk, has been described yet [83] (Table 2).

In conclusion, probiotics and/or symbiotics might not be administered in severe intestinal failure cases. However, clear cases of potentially harmful bacteriemia have not been described.

### 5.2. Promising Effectiveness of Immunonutrition

There is a growing number of recently published research suggesting promising effects of immunonutrition on acute respiratory infections [84,85]. The immune-nutrition definition is: “modulation of either the activity of the immune system or modulation of the consequences of activation of the immune system by nutrients or specific food items fed in amounts above those normally encountered in the diet” [86]. As an example of immunonutrition, there is a formula rich in casein used with success in pediatric IBD populations, showing effectiveness in reducing and maintaining intestinal inflammation [87]. Other examples of immune-nutrition formulations are composed of vitamins (e.g., A, C, D), microelements (e.g., zinc, selenium), omega-3 fatty acids, phytochemicals (e.g., flavonoid, curcumin) [63,64].

They have been used both as preventive measures prior to major surgical interventions and in the recovery phase as oral and non-oral food supplements (e.g., for enteral nutrition) [88,89]. In both cases, they have shown a positive impact on early recovery from ICU condition, reduction of sepsis, SIRS, MOF incidence, and prevention and reduction of malnutrition/sarcopenia development [90,91] (Figure 1).

The targets of immunonutrition are mainly inflammatory cascade and gut microbiota. Immunonutrition approaches stimulate microbial metabolism of SCFA (e.g., acetate, propionate, and butyrate). The latter are crucial in intestinal barrier function because they provide fuel to colonic epithelial cells and promote regulatory T cell (Treg) function. SCFA are also absorbed into the systemic blood circulation and they bind to the G protein-coupled receptors GPR41 and GPR43. These receptors endorse protective immunity [92].

More in detail, the best evidence of the anti-inflammatory activities of SCFA belongs to butyrate. It is able to modulate the maturation of dendritic cells (DC) and increase the number of Treg cells and levels of the anti-inflammatory cytokines like IL-10. On the other hand, butyrate inhibits the production of pro-inflammatory cytokines (namely, interferon-C and IL-2 [93,94]. A healthy diet and, in particular, whey protein rich in casein contained in IBD-designed immuno-nutrition, stimulate the production of the anti-inflammatory transforming growth factor-beta, retinoic acid, and thymic stromal lymphopoietin [89].

These actions on colonocytes and their interactions with immune-tolerance shape gut microbiota composition: enterocytes’ metabolism of butyrate consumes oxygen leading to hypoxia with the promotion of growth of strictly anaerobic bacteria such as *Firmicutes*, resembling “eubiosis” [95].

In recent times, exactly in the time span of the last three years, specific immuno-nutrients have been shown to be effective in the treatment of COVID-19 patients. In detail, immunonutrition is effective for the reduction of innate and adaptive immune responses responsible for the “cytokines’ storm” typical of COVID-19 patients [96]. There is already solid evidence supporting the use of specific protein-rich food nutrient supplements on COVID-19 patients’ morbidity and mortality [97]. In detail, in the literature, some reports evaluated the impact of nutrition in ICU COVID-19 patients with early manifestations of malnutrition and, sarcopenia [98]. It is important to recognize that both malnutrition/sarcopenia and hyper-inflammatory state are significantly associated with morbidity and mortality of COVID-19 patients [99] (Figure 1). The effects of immuno-nutrition approaches on COVID-19 patients can be explained by their anti-inflammatory effects, an increase in SCFA production, a consequent improvement in the metabolism of colonocytes, and the re-shaping of healthy gut microflora [100].

Interestingly, Brazilian researchers investigated the impact of the hyper-proteic normo-caloric diet with or without immune-nutrition formula used as an add-on treatment. The authors found a significant reduction of inflammatory response and related lymphopenia in non-ventilated COVID-19 patients [101]. Our group found similar results on two subsets of COVID-19 patients admitted to ICU and, semi-intensive clinical units under invasive and non-invasive ventilation, respectively. Importantly, the whey-protein rich we used was able to reduce time of ventilation until earlier extubating time, morbidity, and mortality among ventilated ICU patients, with a consensual reduction of inflammatory indexes and prevention of malnutrition development as assessed by bioimpedance vectorial analysis (BIVA) [102]. Subsequently, we studied the impact of the same immune-nutrition formula in semi-intensive COVID-19 patients under non-invasive ventilation. This group of patients, typically from the third wave of the SARS-CoV-2 pandemic, were obese in prevalence, thus with a significant percentage of subjects with sarcopenia. These patients showed a similar reduction of inflammatory response after immunonutrition and prevention of malnutrition development [103] (Figure 1).

The evidence on the role of immuno-nutrition in a hyper-inflammatory disease such as COVID-19 and/or IBD confirms the significant impact of nutrition in the modulation of immune system functioning and maintenance of gut eubiosis, especially when used in the ICU setting. Moreover, this approach seems safe and devoid of detrimental side effects, with a good compliance profile.

## 6. Conclusions and Future Perspectives

Gut eubiosis and its functions in the frame of human health have been deeply studied and elucidated. This has led to important and very promising results in terms of disease prevention and treatment in non-critical patients. In detail, the use of probiotics, FMT, and nutrition in patients is bringing surprising clinical outcomes. The latter were not predictable 30 years ago. That was the time the gut microbiota concept was limited to the intestine and its functions.

A critically ill patient is a peculiar human subject that has a depressed immune-surveillance status and, conversely, a hyper-inflammatory state reacting to the cause of ICU admittance. This results in a severe risk of malnutrition due to protein consumption and serious risks for survival.

ICU patients have a specific form of gut dysbiosis that is a result and, in turn, perpetuates the systemic inflammatory state of these subjects. Thus, it is a therapeutic target for physicians and researchers. The use of probiotics, FMT, and immunonutrition to prevent and/or treat microbial imbalance shows preliminary, small sample-size reports, perhaps enriched by the COVID-19 pandemic.

First, larger multicentric studies, then, standardized gut remodulatory approaches are warranted to confirm these results. A subsequent, closer phase of research is represented by precision medicine describing per-person gut dysbiosis and treating this via the artificial intelligence-based design of personalized probiotics “cocktails”.

## Figures and Tables

**Figure 1 nutrients-15-03569-f001:**
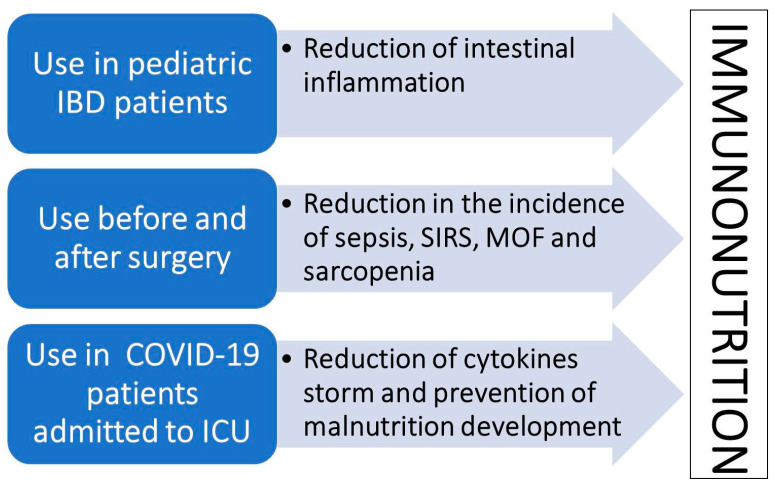
Main effects of immuno-nutrition according to literature evidence.

**Table 1 nutrients-15-03569-t001:** Gut microbiota in critically ill patients.

Non-ICU Patients	ICU Patients
***Phyla***: *Firmicutes*, *Bacteroidetes*, *Proteobacteria*, *Actinobacteria*, *Verucomicrobia*, *Fusobacteria* [11,12,13] ***Genera***: *Clostridium*, *Lactobacillus*, *Enterococcus*, *Streptococcus*, *Bacteroides*, *Prevotella*, *Pseudomonas*, *Acinetobacter*, *Corynebacterium*, *Actinomyces*, *Bifidobacterium*, *Akkermansia*, *Fusobacterium* [11,12,13]	↑↑ ***Phyla***: *Firmicutes*, *Bacteroidetes*, *Proteobacteria*, *Actinobacteria* [25,26,27] ↑↑ ***Genera***: *Clostridium*, *Lactobacillus*, *Enterococcus*, *Streptococcus*, *Bacteroides*, *Prevotella*, *Pseudomonas*, *Acinetobacter*, *Corynebacterium*, *Actinomyces*, *Bifidobacterium* [25,26,27] ↓↓ ***Phyla***: *Verucomicrobia*, *Fusobacteria* [25,26,27] ↓↓ ***Genera***: *Akkermansia*, *Fusobacterium*, *Faecalibacterium* [25,26,27]

Notes. ↑: increased and ↓: reduced abundance and diversity. The table lists the most abundant microbial phyla and related genera from the intestinal microbiota of non-ICU and ICU patients.

**Table 2 nutrients-15-03569-t002:** Clinical trials regarding the use of probiotics in critically ill patients.

References	Patients	Probiotics	Outcome
Hickson et al. [45]	135 hospital patients	*L. casei*, *L. bulgaricus*, and *Streptococcus thermophiles*	Reduction in the incidence of diarrhea of 12% vs. 34% from the control group
D’Souza et al. [46]	Immunocompromised patients from nine different randomized trials	*Saccharomyces boulardii*, *L. rhamnosus* GG, *B. longum*	Prevention of diarrhea outbreaks
Bleichner et al. [47]	128 ICU patients	*S. boulardii*	Reduction in the incidence of diarrhea only in the group treated with probiotics
Sugawara et al. [53]	81 patients subjected to hepatectomy	*Lactobacillus casei*, *Bifidobacterium breve*	Significantly increase of natural killer cell activity, lymphocyte counts, and reduction of pro-inflammatory cytokines levels after hepatectomy vs. control group
Batra et al. [55]	1103 ICU patients from nine different randomized trials	*Lactobacillus* spp., *Pediococcus* spp., *Leuconostoc* spp., *Bifidobacterium* spp., *Bacillus subtilis*, *Streptococcus* spp., *Ergyphilus* spp., *Bifidus* spp., *Saccharomyces* spp., *Enterococcus* spp.	Significant reduction of VAP incidence and mortality
Ceccarelli G. et al. [57]	200 patients with severe COVID-19 pneumonia	*Streptococcus thermophilus*, *Bifidobacterium lactis*, *Lactobacillus acidophilus*, *Lactobacillus helveticus*, *Lactobacillus paracasei*, *Lactobacillus plantarum*, and *Lactobacillus brevis*	Significant association with reduced risk of death
Olah et al. [61]	45 patients with acute pancreatitis	*Lactobacillus plantarum*	Significant reduction of infectious complications

## Data Availability

Not applicable.

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
