# Peer review of "Gut Microbiota and Critically Ill Patients: Immunity and Its Modulation via Probiotics and Immunonutrition"

_nutrients, 2023, doi:10.3390/nu15163569_

Round 1
Reviewer 1 Report
This is an interesting review focusing on gut microbiota composition/ changes and its metabolites in critically ill patients as well as their pathophysiological role and possible modulation through probiotics, FMT and immunonutrition.
Specific comments
- Figure 1 should be completed and clearer : the various T cell subpopulations and their cytokine secretions leading to increased adaptive immune responses and inflammation in critically ill patients should be clarified in a scheme, in relation with dysbiosis.
- Table 1 : the table could be reorganized to make it easier to read.
Faecalibacterium prausnitzii, one of the most abundant commensal bacterial, that plays a key role in the maintenance of gut homeostasis via its anti-inflammatory properties, is not mentioned in this table. F. Prau is dramatically reduced in IBD patients. Has a decrease in F. Prau. been reported in critically ill patients, in addition to other microbial phyla? If yes, this should be mentioned.
-5.1. Probiotics and their derivatives: a table or scheme could be added to sum up all the informations provided in the text
- The paragraph concerning FMT is rather short. If randomized clinical trials using FMT have been reported or are in progress in critically ill patients, this should be mentioned. (and what about FMT in COVID-19 patients?)
Author Response
#Referee 1
This is an interesting review focusing on gut microbiota composition/ changes and its metabolites in critically ill patients as well as their pathophysiological role and possible modulation through probiotics, FMT and immunonutrition.
Specific comments
- Figure 1 should be completed and clearer : the various T cell subpopulations and their cytokine secretions leading to increased adaptive immune responses and inflammation in critically ill patients should be clarified in a scheme, in relation with dysbiosis.
We have upgraded and updated the figure 1.
- Table 1 : the table could be reorganized to make it easier to read.
We thank the reviewer for suggestion. We have modified it accordingly.
Faecalibacterium prausnitzii, one of the most abundant commensal bacterial, that plays a key role in the maintenance of gut homeostasis via its anti-inflammatory properties, is not mentioned in this table. F. Prau is dramatically reduced in IBD patients. Has a decrease in F. Prau. been reported in critically ill patients, in addition to other microbial phyla? If yes, this should be mentioned.
We thank the reviewer for this useful and accurate observation. We have added this to the table.
-5.1. Probiotics and their derivatives: a table or scheme could be added to sum up all the informations provided in the text.
We thank the reviewer for this suggestion. We have added the appropriate table.
- The paragraph concerning FMT is rather short. If randomized clinical trials using FMT have been reported or are in progress in critically ill patients, this should be mentioned. (and what about FMT in COVID-19 patients?).
We thank the reviewer for this interesting suggestion. We have expanded the section.

Reviewer 2 Report
This article summarized the intestinal microbiota dysfunction in critically ill patients, the pathophysiological role of intestinal microbiota disturbance in critically ill patients, and the clinical prospect of probiotics in some critically ill patients. I have found the following problems, and hope to help you improve your paper.
1. As it is said in the passage "Under critical conditions, the composition of gut microbiota is deranged by several factors, also including treatments in use", so how do you distinguish between critical injury that causes dysregulation of the gut microbiota and treatment for critical injury that causes dysregulation of the gut microbiota?
2. In the third part, the original research content on intestinal microecological dysregulation in critically ill patients is too thin and needs to be enriched.
3. About the sentence "Looking at summarizing par-145 adigms, the extreme balance of Bacteroidetes and Firmicutes ratio can be used as a pre-146 dictor of mortality in ICU patients, also describing the imbalanced immunity ", is it supported by references?
4. Is there any direct observation and study on the imbalance between Bacteroidetes and Firmicutes in critically ill patients? If you have, please add to the article to enrich the argument.
5. When discussing the application of probiotics in critically ill patients, you can categorize the discussion by disease type to enhance the organization of the article.
6. Is immunonutrition therapy used for COVID-19 patients related to the regulation of intestinal flora? And what is the significance of your argument about immunonutrition therapy?
To sum up, the logic and evidence of this paper are not good enough to fully discuss the intestinal flora disturbance in critically ill patients and its mediated pathological mechanism. Therefore, I think that this paper needs further improvement and is not suitable for publication in this journal.

Author Response
#Referee 2
This article summarized the intestinal microbiota dysfunction in critically ill patients, the pathophysiological role of intestinal microbiota disturbance in critically ill patients, and the clinical prospect of probiotics in some critically ill patients. I have found the following problems, and hope to help you improve your paper.
- As it is said in the passage "Under critical conditions, the composition of gut microbiota is deranged by several factors, also including treatments in use", so how do you distinguish between critical injury that causes dysregulation of the gut microbiota and treatment for critical injury that causes dysregulation of the gut microbiota?
We thank the reviewer for this interesting question rising a though issue. We have specified the difficulty addressing the exact weight of any factor potentially affecting gut dysbiosis in critical patient and made some example where evidence is available.
- In the third part, the original research content on intestinal microecological dysregulation in critically ill patients is too thin and needs to be enriched.
We thank the reviewer for this suggestion. We have enriched the text according to the comments and suggestions.
- About the sentence "Looking at summarizing par-145 adigms, the extreme balance of Bacteroidetes and Firmicutes ratio can be used as a pre-146 dictor of mortality in ICU patients, also describing the imbalanced immunity ", is it supported by references?
We thank the reviewer for the annotation, we have provided one.
- Is there any direct observation and study on the imbalance between Bacteroidetes and Firmicutes in critically ill patients? If you have, please add to the article to enrich the argument.
We thank the reviewer for the question. Yes, we have added the required evidence in order to enrich the section.
- When discussing the application of probiotics in critically ill patients, you can categorize the discussion by disease type to enhance the organization of the article.
We thank the reviewer for this suggestion that helped to better organize the text. We have followed it accordingly.
- Is immunonutrition therapy used for COVID-19 patients related to the regulation of intestinal flora? And what is the significance of your argument about immunonutrition therapy?
We thank the reviewer for the interesting questions. We have added pathophysiologic evidence on regulation mechanism of gut microflora by immunonutrition and its significance.
To sum up, the logic and evidence of this paper are not good enough to fully discuss the intestinal flora disturbance in critically ill patients and its mediated pathological mechanism. Therefore, I think that this paper needs further improvement and is not suitable for publication in this journal.

Round 2
Reviewer 2 Report
The authors have refined the questions I raised earlier. And this paper can be published with a polished expression of language.
The language of this paper needs further polishing.
Author Response
Dear Referee,
thank you for your positive evaluation. As suggested, we perform the language revision of the text. The changes are highlithed in yellow
Kind regards